# Determinants of Prenatal Childbirth Fear during the Third Trimester among Low-Risk Expectant Mothers: A Cross-Sectional Study

**DOI:** 10.3390/healthcare12010050

**Published:** 2023-12-25

**Authors:** Heba A. Ibrahim, Majed S. Alshahrani, Wafaa T. Ibrahim Elgzar

**Affiliations:** 1Nursing College, Najran University, Najran 66261, Saudi Arabia; 2College of Medicine, Najran University, Najran 66261, Saudi Arabia

**Keywords:** fear of childbirth, social support, parturition, prevalence, pregnant women, Saudi Arabia

## Abstract

Background: Fear of Childbirth (FOC) can significantly impact women’s physical and psychological health; therefore, healthcare providers must provide proactive care, which means they have to intervene before FOC becomes tokophobia. This study’s purpose is to evaluate the determinants of prenatal childbirth fear during the third trimester among low-risk expectant mothers. Methods: A cross-sectional comparative study was conducted at the Maternal and Children Hospital’s outpatient clinics in Najran City, Saudi Arabia, from April to July 2023. The study involved 377 nulliparous and multiparous women, using a systemic random sampling technique. The data were collected using an interview schedule composed of questions related to demographic and obstetrics characteristics, the FOC questionnaire, and a multidimensional scale of perceived social support. Significant FOC predictors were examined using a binary logistic regression model. Results: There was a statistically significant difference between nulliparous and multiparous participants concerning FOC; 80.0% of nulliparous participants had significant FOC compared to 67.8% of multiparous participants (*p* = 0.011). A binary logistic regression clarified that regular antenatal care and family and spousal support were significantly negatively correlated with significant FOC among multiparous and nulliparous women (*p* < 0.05). For multiparas, FOC was associated with pregnancy planning and previous labor-related complications. In addition, friends’ support was an important predictor of significantly lower FOC among nulliparous women (*p* < 0.05). Conclusions: Significant FOC was higher in nulliparous women when compared to multiparous women. Numerous obstetric variables and different types of social support play important roles in significant FOC. Special attention and support should be provided to high-risk women for proper FOC management during prenatal classes to improve their childbirth experiences.

## 1. Introduction

### 1.1. Childbirth: Psychological and Physiological Aspects

Childbirth is a unique and complex physical and psychological experience for each woman, one which is accompanied by happiness, anxiety, and fear. Although childbirth is a normal physiological experience, it is considered a woman’s bridge from girlhood to motherhood, with short- and long-term physical, psychological, and social consequences [1]. A stressful childbirth experience is strongly related to postpartum traumatic stress and depression [2].

During the postpartum period, most women tell stories regarding the childbirth experience, a practice that contains physical and psychological dimensions. From the physiologic point of view, there are unique hormonal interactions between a mother and fetus that begin in pregnancy, continue during the postpartum period, and play an important role in labor physiology. Throughout the childbirth process and during the postpartum period, both the maternal and fetal brains are deeply immersed in neurohormonal scenarios that could never be generated artificially [3]. In addition, the woman’s brain secretes oxytocin and endogenous endorphins in an attempt to achieve a balance between the progress of labor and maternal tolerance for pain. Numerous physical, psychological, and cultural factors can influence a woman’s childbirth experience. The most ambiguous and important factor is the Fear of Childbirth (FOC), which can be severe in some cases, called tokophobia, and requires psychiatric intervention. Therefore, there is a growing interest in FOC all over the world [4].

### 1.2. Fear of Childbirth: Prevalence, Components, and Contributing Factors

FOC is a broad concept, and no standardized definition has been determined; therefore, its measurement tools also greatly vary [5,6]. From a psychological point of view, any woman may have some FOC as a difficult human experience. However, the standard is to what extent this fear can influence her ability to cope with the birth process. Demšar et al. conducted a study to explore the prevalence and associated risk factors for FOC among pregnant women. He found that three-quarters of his participants experienced low-to-moderate FOC, while the remaining quarter experienced high or very high FOC [4]. Generally, occurrence of the severe form of FOC, or tokophobia, ranged from 4.8 to 14.8% [5,7]. There are no available references regarding the prevalence of FOC in Saudi Arabia. 

According to recent studies, FOC directly influences psychological distress, and decreasing it can significantly enhance the childbirth experience. In addition, promoting resilience and early screening for FOC may be useful targets for decreasing labor-related psychological distress among pregnant women [8]. Healthcare providers should provide proactive care for FOC, which means they should intervene before FOC becomes tokophobia. When a woman can cope with FOC effectively, she initiates motherhood with a positive experience and a feeling of happiness and satisfaction with the birth of her baby. She also will be able to engage in positive postpartum practices in collaboration with healthcare providers [9]. Therefore, the current study created a new concept of significant FOC; in other words, FOC requires attention and management. 

The components of FOC are numerous and vary from one woman to another. Some women are afraid of the labor process, the probability of emergencies or unpredicted events, and the inability to be involved in decision making. Fear also comes from the possibility of severe labor pain and the body’s ability to control it without self-harm, in addition to fear of clinical procedures such as episiotomy, fear of infant harm, loneliness, and loss of control. Another woman may have a great fear of the unknown, as the whole birth process is stressful [10,11]. FOC can also affect women’s decisions regarding the mode of delivery. 

A recent study explored the role of childbirth in choosing the mode of delivery and concluded that a high level of FOC can be related to women’s unjustifiable requests for cesarean section. They further elaborated that increased fear of harming or distressing the infant, fear from pain, fear from the body’s ability to give birth, and fear of not being involved in decision making seem to be significant dimensions of childbirth fear associated with cesarean section preference [12].

FOC is mostly expressed verbally by women, though some women may experience nightmares, psychosomatic symptoms, an inability to concentrate on work tasks and family activities, and a strong preference for cesarean section [13].

The experience of childbirth is unique to each woman and greatly influenced by numerous factors. Some previous studies reported that FOC was associated with education, age, gestational age, pregnancy stress, childbirth self-efficacy, low social support, and parity [14]. Previous studies found that FOC is more common among nulliparous women than multiparous who exhibit high FOC and tokophobia, with more concern regarding perineal tears and labor pain [15,16]. However, other studies reported a higher incidence among multiparous women with previous experience of birth trauma, as any previous experience exaggerated more fear and uncertainty [17,18]. Therefore, there is still a huge debate regarding the relation between parity at FOC and the associated factors that may contribute to it in multiparous compared to primiparous women. A nulliparous woman may be afraid of unknown, unpredictable events, labor pain, and inability to control the experience, while multiparous women’s FOC may be related to previous negative birth experiences [19]. In addition, social support plays an important role in decreasing pregnant women’s FOC, especially when received from a spouse. Therefore, the family should be involved in FOC management and counseling. Moreover, there is a need to explore if social support differs among nulliparous compared to multiparous women and if it is associated with FOC between the two groups [20]. FOC is important to be managed early during pregnancy, and the first step in its management is to determine its prevalence and associated risk factors in different cultures, as FOC is greatly affected by cultures and beliefs. No available Saudi studies evaluated FOC prevalence and associated factors among nulliparous and multiparous women as a preliminary step to managing such an important problem. In addition, the role of social support in FOC is not explored at the national level and is still debated at the international level. Therefore, the current study aims to explore the determinants of FOC during the third trimester among low-risk expectant mothers.

## 2. Material and Methods

### 2.1. Operational Definition

Significant FOC is a type of FOC that requires medical help to prevent potential drawbacks. In the current study, this variable was assessed using an FOC questionnaire, where a higher total scale score (31 to 60) indicates the presence of significant FOC.

### 2.2. Study Setting

A cross-sectional comparative study was conducted at Maternal and Children Hospital (MCH) in Najran City, Saudi Arabia. MCH is the only governmental hospital that provides maternal and children health services in Najran region; therefore, it serves a large population. The data were collected from four outpatient clinics that provide antenatal services for women with normal, low-risk pregnancies. 

### 2.3. Participants

The participant’s inclusion criteria were age 18–35 years (expected age for safe pregnancy) [21], having normal, singleton, low-risk pregnancy, plan for normal vaginal delivery, gestational age of 28 weeks or more based on the last menstrual period or ultrasound examination from 8 to 12 weeks of gestation (where FOC is most common), parity ≤4 for multiparous women, and willing to participate in the study. Those who had ongoing pregnancy-related complications, diagnosed mental illness, history of stillbirth, or intrauterine death (according to the woman’s medical record) were not eligible to participate in the study.

### 2.4. Sample Size

The sample size was calculated according to the Cocharane formula [22].
$$n=\frac{Z^{2}P\left(1-P\right)}{d^{2}}$$
where *Z* = 1.96 for a 95% confidence interval; *P* = prevalence of FOC from a previous study (73%) [11] among women in the third trimester of pregnancy, and *d* is the margin error (0.05). The minimum sample size was 302 pregnant women; after adding 15% to compensate for the anticipated sample loss, the total sample size was 350 women. 

### 2.5. Sampling Technique

The participants were recruited from outpatient clinics using a systemic random sampling technique. The data-collection team was composed of two researchers and two data collectors. Each data collector expressed the ability to collect 5 cases daily; therefore, 20 cases could be collected daily. Each one of the four antenatal clinics served around 15 cases daily based on the clinic registry system; the total number of cases served by the four clinics daily was around 60. The sampling interval was calculated by dividing the total flow rate of the 4 clinics (60) by the total cases expected to be collected daily (20), and it was 3. A random starting point was picked from 1, 2, or 3, and the sampling interval was maintained. The cases were picked up from the clinic waiting areas, and if one of the recruited cases did not meet the inclusion criteria, she was replaced by the next one, and the sampling interval was maintained.

### 2.6. Data-Collection Tools

The data were collected using an interview schedule composed of four main parts. Part I was concerned with collecting data related to demographic characteristics such as age, residence, occupation, education, and satisfaction with family income. Part II was concerned with obstetric history and collection of data such as gravidity, parity, gestational age, number of abortions, and living children. It also includes data on previous labor complications, planned or unplanned pregnancies, and regularity of antenatal care.

Part III, the FOC questionnaire, was utilized to evaluate the pregnant women’s perceived FOC. It was developed by Slade et al. [23] and incorporates 20 items rated on a 4-point Likert scale: strongly disagree (0), disagree (1), agree (2), and strongly agree (3). The scoring system was reversed in item numbers 1, 3, 5, 8, 10, 14, 17, and 20. The scale was designed to evaluate ten dimensions related to FOC, namely, fear of not being able to know and plan for unpredictable events, fear of harming or distressing the infant, fear of pain, fear of the body’s ability to give birth, fear of self-harm during delivery or postnatal period, fear of clinical procedures, fear of not being involved in decision making, fear of loneliness, fear of the loss of control, and fear of unknown. The total scale score ranged from 0 to 60, with a higher score indicating higher fear. Non-significant fear is considered for values from 0 to 30, and significant fear is considered for values from 31 to 60 [20]. The total scale revealed high reliability (r = 0.84), as reported by Sanjari et al. [24]. 

Part IV is a Multidimensional Scale of Perceived Social Support (MSPSS). This scale was developed by Zimet et al. [25] to evaluate the social support provided by the family (4 items), friends (4 items), and significant others (4 items). The total scale was composed of 12 items rated on a 5-point Likert scale ranging from strongly disagree (1) to strongly agree (5). The total scale score ranged from 12 to 60, with higher scores indicating higher social support. The Cronbach’s α coefficient for family, friends, and significant other subscales were 0.87, 0.85, and 0.91, respectively. The total scale reliability was 0.88, as reported by Zimet et al. [25]. 

### 2.7. Data-Collection Methods

After obtaining the necessary approvals, the participants were recruited according to the previously mentioned sampling procedures. Data collection was performed in the clinics’ waiting areas from April to July 2023 in both morning and afternoon shifts. For each recruited case, the data collector explained the study purpose and obtained informed consent from the participant. The researchers ensured the inclusion criteria and then interviewed the pregnant woman to complete the interview schedule. If the woman did not meet the inclusion criteria or refused participation, she was replaced by the next one while maintaining the sampling interval. After the interview, the woman was allowed to ask questions, and the researchers provided complete answers. 

The flowing flow chart illustrates the participants’ distribution (Figure 1).

### 2.8. Data Quality Control

Two of the data collectors were researchers with previous experience in data collection. The remaining two data collectors were bachelor’s degree holders with previous experience in data collection. Before data collection, two meetings were provided to the data collectors to explain the study proposal, interview schedule, and research ethics.

### 2.9. Ethical Approvals

The Dean of Scientific Research at Najran University approved the research proposal. In addition, the institutional review board of Najran Health Affairs examined the study proposal and data-collection tools and approved the study (IRB Log Number 2023-07 E). Later, the MCH hospital administration was notified before data collection. At the beginning of each interview, the study purpose was explained, data confidentiality was ensured, and anonymity was clarified. The woman was also informed about her right to refuse participation without any consequences regarding the care provided. Informed consent was taken, and the woman was informed about her right to see study results after publication.

### 2.10. Statistical Analysis

After completing data collection, the data were entered into IBM version 23. Descriptive statistics were used to describe data, such as number, percentage, mean, and standard deviation (SD). The study’s dependent variable, FOC, was numeric, while the independent variables were numeric and categorical. The numeric variables were age, gestational age, number of abortions, living children, and multidimensional social support. The categorical variables included education, residence, gravidity, satisfaction with income, history of previous labor-related complications, regularity of antenatal care, and planning for the current pregnancy. The total FOC, social support scale, and subscales were obtained by summing items. Differences between nulliparous and multiparous women were checked using chi-square, Fisher exact, and independent *t*-tests. A binary logistic regression was used to examine the associated factors of significant FOC, and an adjusted odd ratio (AOR) was calculated with a 95% Confidence Interval (CI). The significance level was considered at *p* < 0.05, and multicollinearity was checked before binary logistic regression. The Cox & Snell R Square goodness of fit test checked the final regression model.

## 3. Results

### 3.1. Participants’ Socio-Demographic and Obstetrics Characteristics

In the current study, 52.5% of participants were multiparous, while 47.5% of the whole sample were nulliparous. The participants’ mean age was 26.53 years, with the majority (85.2%) being residents of urban areas. Among them, approximately half (50.1%) were housewives, and over three-quarters (80.1%) had received higher education. The majority had a planned pregnancy and no previous labor-related complications, 94.4% and 91.5%, respectively, and around three-quarters (73.9%) reported regular antenatal care. In addition, the mean gestational age was 31.28 weeks, and the mean number of abortions was 0.29. By comparing socio-demographic and obstetric characteristics among nulliparous and multiparous participants, the findings indicated significant differences in age, occupational status, pregnancy condition, and number of abortions. However, there were no statistically significant differences regarding residence, education, monthly income, antenatal care, and gestational age, as shown in Table 1.

### 3.2. FOC and MSPSS Scores among Participants

The results of the *t*-test revealed statistically significant differences (*p* < 0.05) between nulliparous and multiparous participants in experiencing fear of harming or distressing the infant, fear of pain, fear of the body’s ability to give birth, fear of clinical procedures, fear of not being involved in decision making, fear of loneliness, fear of the loss of control, fear of the unknown, and overall FOC score. There were no statistically significant differences (*p* > 0.05) in family, friends, spousal support, and overall MSPSS scores between the nulliparous and multiparous participants (Table 2).

Figure 2 illustrates the prevalence of FOC among nulliparous and multiparous participants. There was a statistically significant difference between nulliparous and multiparous participants in experiencing FOC. More than three-quarters (80%) of nulliparous participants had significant FOC compared to 67.8% of multiparous participants (χ^2^ = 6.439, *p* = 0.011).

### 3.3. Determinant of FOC

As shown in Table 3, a binary logistic regression clarified that antenatal care, family, and spousal support were predictors for significant FOC among multiparous and nulliparous participants. Planning for pregnancy and previous labor-related complications were associated with significant FOC among multiparous participants, while friend support was associated with significant FOC among nulliparous participants. Multiparous women who had an unplanned pregnancy had a higher probability of experiencing significant FOC than those who had planned pregnancy (AOR = 5.360, *p* = 0.040)). Multiparous and nulliparous women who had irregular antenatal care had a higher probability of experiencing significant FOC than those with regular antenatal care (AOR = 1.857, *p* = 0.035) and (AOR= 2.607, *p* = 0.012), respectively. Multiparous women with previous labor-related complications had a higher probability of having FOC, by 5 times (AOR = 5.605, *p* = 0.005), compared to those without previous labor-related complications. Furthermore, multiparous women with a high family and spousal support had a lower probability of having significant FOC (AOR = 0.835, *p* = 0.012 and AOR = 0.765, *p* < 0.001, respectively). Similarly, nulliparous women with high family, friend, and spousal support were less likely to have significant FOC (AOR = 0.681, *p* = 0.001; AOR=0.683, *p* = 0.002; and AOR= 0.608, *p* = 0.001, respectively).

## 4. Discussion

### 4.1. FOC Prevalence among Nulliparous and Multiparous Women

This study examined the differences in nulliparous and multiparous women’s FOC. Our findings indicated that nulliparous women had a higher prevalence (80%) of significant FOC than multiparous women (67.8%). In addition, when comparing the mean FOC score, it was higher in nulliparous participants when compared to multiparous women, with a statistically significant difference in FOC between the two groups. The current findings agree with several prior studies [26,27,28,29,30] that explained the differences in FOC among nulliparous and multiparous women. They indicated that nulliparous women were more likely to have a higher FOC. Other studies reported a higher FOC among nulliparous women compared to multiparous women; however, the differences between the two groups were not statistically significant [31,32]. It is believed that multiparas have prior childbirth experience; therefore, they may be better equipped for subsequent childbirth and have a lower risk of FOC than nulliparous women. However, previous negative childbirth experiences, such as dystocia, an advanced degree of perineal tears and lacerations, and assisted vaginal deliveries, may increase the risk of FOC [33]. Thus, multiparas with a negative childbirth experience may have higher levels of FOC than nulliparous women due to previous traumatic childbirth and suffering from post-traumatic stress disorder [19]. Therefore, nurses and midwives should remember and consider the difference between nulliparous and multiparas concerning FOC. Consequently, special attention should be given to nulliparous women and those with previous negative childbirth experiences to properly manage FOC during prenatal classes and improve their childbirth experiences.

In the current study, nulliparous and multiparous women were different in terms of their FOC key elements. Fear of harming or distressing the infant, fear of pain, fear of the body’s ability to give birth, fear of clinical procedures, fear of not being involved in decision making, fear of loneliness, fear of the loss of control, and fear of the unknown were higher in nulliparous than multiparous participants, with statistically significant differences between the two groups. In this regard, no studies examine the differences in the main components of FOC between nulliparous and multiparous women. However, Sheen and Slade [34] conducted a meta-synthesis and reported that six main components of FOC were common among women. These components were fear of the unknown, the possibility of injury, pain, inability to give birth, loss of control, and adequacy of support from caregivers. Sheen and Slade did not compare the different components of FOC among nulliparous and multiparous women, but they identified the same concern expressed by the current study participants, which means that the human experience of FOC is common and similar across cultures and regions.

Furthermore, Slade et al. [35] tried to explore FOC from different perspectives as they included both women and midwives in their study’s participants. They reported that women and midwives identified seven themes for FOC, which were also similar to what was reported by the current study. Namely, the seven FOC components were fear of the unknown and not being able to plan for unpredictable events, fear of hurting or stressing the baby, fear of not being able to deal with pain, fear of harming oneself during labor and after birth, fear of being exposed, fear of not having a voice in decision making, and fear of abandonment and loneliness. Additionally, in the Saudi context, a very recent study conducted by Elgzar et al. [12] explained the significant component of FOC associated with cesarean section preference. They elaborated that fear of harming or distressing the infant, fear of pain, fear of the body’s ability to give birth, and fear of not being involved in decision making were associated with cesarean section preference. In general, the FOC components seem to be similar across cultures, regions, and ethnicities, as the process of labor and the psychology of childbirth are unified human experiences with similar components but different associated factors. The results of the current study could provide significant insights into women’s experiences of childbirth and the emotional elements of childbirth. Consequently, understanding these experiences could be used to assess and improve reproductive health policies and services.

### 4.2. Determinants of FOC among Nulliparous and Multiparous Women

The current study’s findings revealed that multiparous women with unplanned pregnancies had a higher probability of experiencing significant FOC than those with planned pregnancies. Planning for pregnancy was not a significant predictor of FOC among primiparous women, as most of them already had planned pregnancies. This finding is supported by other international studies conducted in China [36], Turkey [37], and southern Ethiopia [31]. The reason behind this result may be attributed to the increased stress and anxiety among women with unplanned pregnancies, besides physiological adjustment during pregnancy, which increases the burden on their bodies. An unplanned pregnancy indicates that the multigravida is more likely to have unexpected and stressful life events, which may negatively affect her coexisting life plan. These sudden and unplanned circumstances may lead to increased FOC among this group [31]. In this regard, a recent study in France indicated that a gravida with unplanned pregnancy had an increased risk of maternal psychological distress, maladaptation with pregnancy, and a higher risk of postpartum depression [38].

The results of this study expressed a great association between irregular antenatal follow-up and a significant level of FOC among both primiparous and multiparous women. These results agree with similar studies undertaken in Kenya [29] and Ethiopia [39]. They reported that pregnant women who had a regular follow-up of their pregnancy were less likely to have FOC than those who did not. In fact, during an antenatal check-up, the gravida woman expresses her FOC to their medical care providers; therefore, the necessary information is clarified to reassure the woman and eliminate unnecessary fears. In addition, healthcare providers play an important role in reducing FOC through prenatal classes and education. The gravida woman who received antenatal psychoeducation reported experiencing lesser FOC than those who did not receive it in a randomized controlled trial conducted in Australia [7].

Previous labor-related complications are other essential factors associated with FOC among multiparous women in the current study. Women with previous labor-related complications were more likely to have a significant FOC than those without. This finding aligns with studies conducted in Eastern Ethiopia [39] and Norway [40]. The possible explanation for the current result is that FOC may be exaggerated by a woman’s prior experience with stress and insufficient counseling provided by health care providers. The most common sources of FOC for women with previous labor-related complications were the previous experience of severe pain, harming or distressing the infant, cesarean section, premature birth, prolonged labor, and postpartum complications [41]. Another Chinese study also reported a positive association between FOC and prior experience of miscarriage [42]. Policies to determine and assist pregnant women with bad childbirth experiences and previous obstetrics complications may be useful. Such women may also require additional care and support during the postpartum period to control the FOC during the postpartum period and subsequent pregnancies.

### 4.3. The Role of Social Support in FOC

Additionally, our findings indicated that primiparous and multiparous women with good family and spousal support were less likely to have FOC. In addition, nulliparous women with low friend support had a higher probability of significant FOC. At the same time, multiparous women’s FOC was not affected by the degree of friend support. This finding aligns with two Chinese studies [14,36], a study conducted in six European countries [43], and an Ethiopian study [44]. The former studies emphasized the importance of social support in decreasing women’s stress and anxiety during the late pregnancy and childbirth process. In fact, strong support from the whole family, especially the spouse, can reinforce women’s confidence that birth is a physiological process, thus leading to psychological well-being and declining FOC. No available research in the database explored the role of friendship in FOC. However, primigravids’ age is mostly relatively younger than multigravidas; therefore, they may be strongly affected by friendship. Primiparous women with strong social networks, especially from friends, are expected to cope with pregnancy anxiety and FOC more effectively. However, further studies are needed to explore the role of friendship in the FOC. Finally, the current study addressed important and ignored childbirth-related problems in Saudi Arabia. Our study clarified important modifiable associated factors of FOC, which, if probably managed, could lead to a more positive birth experience.

### 4.4. Strengths and Limitations of the Study

Our study has numerous strengths. It is the first study conducted in Saudi Arabia to evaluate the prevalence, associated factors, and the role of social support in significant FOC among nulliparous and multiparous women. Our study used recent, reliable, and valid instruments to evaluate FOC and social support. In addition, the systemic random sampling technique provides sufficient power to analyze the roles of various FOC-associated factors. Although the current study used standardized, validated scales in data collection, there is a risk of self-reported bias. In addition, the current study did not evaluate some FOC-associated factors, such as self-efficacy and the mode of delivery preference. Although the current cross-sectional study investigated the determinants of FOC among nulliparous and multiparous women, a cause–effect relationship cannot be established.

### 4.5. Practical Implications

The current study findings can help healthcare providers identify high-risk women for FOC, with special emphasis on nulliparous women. Consequently, specialty antenatal educational clinics, which are managed by midwives who are interested in FOC and serve women using a personalized approach, should be established. The specialty antenatal educational clinics’ team may involve an interdisciplinary team of midwives, obstetricians, social workers, and psychologists, as appropriate, for each woman.

### 4.6. Further Studies

Further studies are needed to explore the role of friendship, self-efficacy, and mode of delivery preference in FOC.

## 5. Conclusions

A significant FOC was higher in nulliparous when compared to multiparous women. Numerous obstetric variables and different types of social support play an important role in significant FOC. Special attention and support should be provided to high-risk women for proper FOC management during prenatal classes to improve their childbirth experiences.

## Figures and Tables

**Figure 1 healthcare-12-00050-f001:**
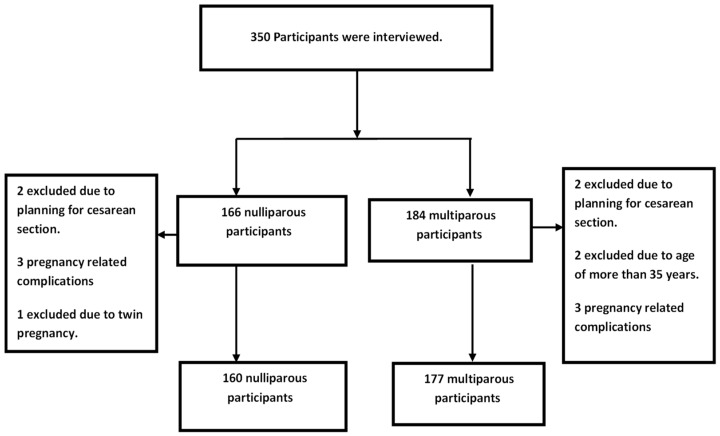
Participants’ flow chart.

**Figure 2 healthcare-12-00050-f002:**
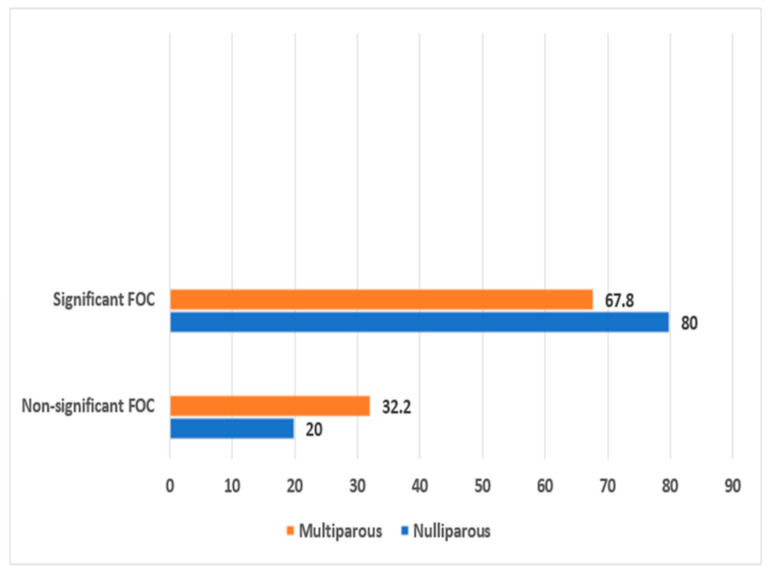
The Prevalence of childbirth fear among nulliparous and multiparous participants.

**Table 1 healthcare-12-00050-t001:** Socio-demographic and obstetrics characteristics among nulliparous and multiparous participants (*n* = 337).

Variables	Total Sample *n* = 337	Parity		X^2^/FET/*t*-Test	*p*
		Nulliparous *n* = 160	Multiparous *n* = 177		
	*n* (%)	*n* (%)	*n* (%)		
**Age (years) (M ± SD)**	**26.53 ± 6.50**	24.08 ± 4.95	28.74 ± 6.94	7.034	<0.001 **
**Residence**				1.317	0.251
– Rural	50 (14.8)	20 (12.5)	30 (16.9)		
– Urban	287 (85.2)	140 (87.5)	147 (83.1)		
**Occupational status**				6.586	0.010 *
– Housewife	169 (50.1)	92 (57.5)	77 (43.5)		
– Employee	168 (49.9)	68 (42.5)	100 (56.5)		
**Education**				3.047	0.218
– Read and write	23 (6.8)	8 (5.0)	15 (8.5)		
– Secondary education	44 (13.1)	25 (15.6)	19 (10.7)		
– Higher education	270 (80.1)	127 (79.4)	143 (80.8)		
**Monthly income**				0.396	0.820
– Insufficient	21 (6.2)	11 (6.9)	10 (5.6)		
– Sufficient	96 (28.5)	47 (29.4)	49 (27.7)		
– Sufficient and saving	220 (65.3)	102 (63.8)	118 (66.7)		
**Planning for pregnancy**				14.390	<0.001 **
– Planned	318 (94.4)	159 (99.4)	159 (89.8)		
– Unplanned	19 (5.6)	1 (0.6)	18 (10.2)		
**Antenatal care**				3.215	0.073
– Regular	249 (73.9)	111 (69.4)	138 (78.0)		
– Irregular	88 (26.1)	49 (30.6)	39 (22.0)		
**Previous labor-related complications**				NA	NA
– No	162 (91.5)	NA	162 (91.5)		
– Yes	15 (8.5)	NA	15 (8.5)		
**Gestational age (weeks)** **(M ± SD)**	31.28 ± 2.54	31.10 ± 2.65	31.45 ± 2.43	1.269	0.205
**Number of abortions (times)** **(M ± SD)**	0.29 ± 0.73	0.09 ± 0.63	0.44 ± 0.90	3.918	<0.001 **
**Number of parities (times)** **(M ± SD)**	2.1 ± 0.79	NA	2.1 ± 0.79	NA	NA
**Number of living children (M ± SD)**	2.08 ± 1.02	NA	2.08 ± 1.02	NA	NA

Note: M ± SD: Mean ± standard deviation. NA: Not applicable. X^2^: Chi-square test. FET: Fisher’s Exact Test. *t*: Independent sample *t*-test. * Significant at *p* ˂ 0.05. ** Significant at *p* ˂ 0.001.

**Table 2 healthcare-12-00050-t002:** Differences in FOC and MSPSS scores among nulliparous and multiparous participants.

Variables	Study Participants		*t*	*p*
	Nulliparous	Multiparous		
	M ± SD	M ± SD		
**FOC**				
- Fear of not being able to know and plan for unpredictable events	5.42 ± 1.28	5.16 ± 1.23	1.866	0.063
- Fear of harming or distressing the infant	2.86 ± 1.71	2.45 ± 1.34	2.494	0.013 *
- Fear of pain	5.28 ± 1.13	4.85 ± 1.44	2.959	0.003 *
- Fear of the body’s ability to give birth	3.05 ± 1.33	2.78 ± 1.12	1.979	0.049 *
- Fear of self-harm, intra-natal or postnatal	5.36 ± 1.38	5.16 ± 1.24	1.431	0.153
- Fear of clinical procedures	5.44 ± 1.08	5.18 ± 1.17	2.128	0.034 *
- Fear of not being involved in decision making	3.75 ± 1.21	3.44 ± 1.171	2.424	0.016 *
- Fear of loneliness	3.86 ± 1.21	3.44 ± 1.23	3.200	0.002 *
- Fear of loss of control	3.93 ± 1.17	3.63 ±1.11	2.397	0.017 *
- Fear of the unknown	3.91 ± 1.12	3.51 ± 1.08	3.322	0.001 *
**Overall FOC**	42.91 ± 7.72	39.64 ± 6.89	4.104	<0.001 **
**MSPSS**				
- Family support	14.05 ± 1.89	14.44 ± 1.98	1.868	0.063
- Friends support	13.33 ± 1.26	13.20 ± 0.96	1.049	0.295
- Significant other (spouse) support	13.89 ± 1.88	14.10 ± 1.99	1.006	0.315
**Overall MSPSS**	41.27 ± 3.11	41.75 ± 3.47	1.336	0.182

FOC: Fear of childbirth. MSPSS: Multidimensional Scale of Perceived Social Support. M ± SD: Mean ± standard deviation. *t*: Independent sample *t*-test. * Significant at *p* ˂ 0.05. ** Significant at *p* ˂ 0.001.

**Table 3 healthcare-12-00050-t003:** Binary logistic regression analysis of the significant FOC-associated factors.

Associated Factors	Significant Childbirth Fear in Multiparous Women		Significant Childbirth Fear in Nulliparous Women	
	AOR (95% CI)	*p*	AOR (95% CI)	*p*
**Age in years**	0.975 (0.927–1.026)	0.330	1.022 (0.880−1.188)	0.772
**Residence**				
- Rural	Ref			
- Urban	1.176 (0.512–2.700)	0.702	6.590 (0.622−69.767)	0.117
**Occupational status**				
- Housewife	Ref			
- Employee	1.642 (0.939–2.871)	0.082	2.917 (0.910−9.350)	0.072
**Education**		0.184		
- Read and write	Ref			0.679
- Secondary education	4.119 (0.909−18.666)	0.066	1.457 (0.578 −3.674)	0.425
- Higher education	1.023 (0.449−2.33)	0.957	0.743 (0.045−12.160)	0.835
**Monthly income**		0.417		0.523
- Insufficient	Ref	0.648	Ref	
- Sufficient	2.921 (0.569−14.992)	0.199	4.336 (0.330−57.041)	0.264
- Sufficient and saving	1.153 (0.626−2.127)		0.914 (0.272−3.074)	0.885
**Planning for pregnancy**				
- Planned	Ref			
- Unplanned	5.360 (1.081−26.569)	0.040 *	1.113 (0.658−1.884)	0.690
**Antenatal care**				
- Regular	Ref			
- Irregular	1.857 (1.045−3.298)	0.035 *	2.607 (1.229−5.528)	0.012 *
**Previous labor-related complications**				
- No	Ref		-	-
- Yes	5.605 (1.691−18.572)	0.005 *	-	-
**Gestational age in weeks**	1.098 (0.978−1.233)	0.113	1.182 (0.282−4.956)	0.819
**Number of abortions**	1.325 (0.850−2.067)	0.215	1.003 (0.208−4.844)	0.997
**Number of living children**	1.007 (0.734−1.381)	0.966	-	-
**Family support**	0.835 (0.725−0.961)	0.012 *	0.681 (0.540−0.860)	0.001 *
**Friends support**	0.926 (0.732−1.171)	0.523	0.683 (0.535−0.872)	0.002 *
**Significant other (spouse) support**	0.765 (0.666 −0.878)	<0.001 **	0.608 (0.450−0.821)	0.001 *

Note: AOR: Adjusted Odds Ratio. CI: Confidence interval. * Significant at *p* ˂ 0.05 ** Significant at *p* ˂ 0.001.

## Data Availability

The datasets used during the current investigation are available from the corresponding author upon reasonable request.

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
