# Peer review of "Determinants of Prenatal Childbirth Fear during the Third Trimester among Low-Risk Expectant Mothers: A Cross-Sectional Study"

_healthcare, 2023, doi:10.3390/healthcare12010050_

Round 1

Reviewer 1 Report

Comments and Suggestions for Authors

Dear Authors 

The article "Determinant of fear of prenatal birth during the third trimester among low-risk pregnant women" is of interest for clinical practice since, according to the authors, there are no studies on this issue in Saudi Arabia. Fear of childbirth influences the emotional experience of pregnancy and childbirth and can have an impact on maternal and neonatal outcomes, including the type of birth.

The organization of the study was done correctly, and the methodology adequately evaluated the proposed objectives. The results were well systematized and discussed by joint agreement, seeking to respond to the initial objectives. However, the results confirm other studies already carried out without bringing much new information.

As suggested below, some details could be improved to increase the manuscript's clarity.

Title: Understandable, concise and portrays the content of the work. It is consistent with the objectives.

Abstract: The abstract is presented in a structured format and describes the objective(s), method, main results and conclusions

Introduction: Presents the contextualization of the topic in a logical sequence. Explain the research problem's importance and justify the need to carry out the study. The contextualization of the problem can be improved using more current literature.

Methodology: The psychometric characteristics of the scales used in this sample should be included. Clarify the scales used have been validated for the population of Saudi Arabia?

Results: The results are consistent with the study objectives and the method. The data presented is sufficient and allows the authors to develop the proposed analysis.

Discussion: The results were discussed with other authors, highlighting similarities and discrepancies regarding the knowledge already published in the international literature. However, the types of fears and their implications could be explored in greater detail, not just the differences between nulliparous and multiparous women.

It would also be interesting if the authors could explain the implications of the results for clinical practice more clearly. In particular, it identifies strategies for adequately managing fear of childbirth and anticipating adverse outcomes.

Conlusion: They should also consider revising the findings to contribute more meaning to existing knowledge in the area.

References: Authors should update some of the references used in the manuscript to include more recent research. About half of the references are more than five years old.

Author Response

Reviewer  1 comments

We are very grateful to you for sending us your comments on our manuscript; according to these constructive comments provided by you, we have carefully revised the article as described below. They have significantly improved the quality and value of our manuscript. We hope you will find it to be a high-quality scientific work compatible with the standards of the healthcare journal. Please note all reviewer (1) comments are highlighted in blue. The revisions are as follows: 

Comment

Authors response

Page

Paragraph number

The article "Determinant of fear of prenatal birth during the third trimester among low-risk pregnant women" is of interest for clinical practice since, according to the authors, there are no studies on this issue in Saudi Arabia. Fear of childbirth influences the emotional experience of pregnancy and childbirth and can have an impact on maternal and neonatal outcomes, including the type of birth.

Thank you very much for your comments and suggestions to improve our manuscript.

-

-

The organization of the study was done correctly, and the methodology adequately evaluated the proposed objectives. The results were well systematized and discussed by joint agreement, seeking to respond to the initial objectives. However, the results confirm other studies already carried out without bringing much new information.

Thank you very much for your comments and suggestions to improve our manuscript. We considered them in the revised version.

-

-

As suggested below, some details could be improved to increase the manuscript's clarity.

Title: Understandable, concise and portrays the content of the work. It is consistent with the objectives.

Thank you very much for your comments and suggestions to improve our manuscript. We considered them in the revised version.

-

-

Abstract: The abstract is presented in a structured format and describes the objective(s), method, main results and conclusions

Thank you very much for your comments and suggestions to improve our manuscript. We considered them in the revised version.

-

-

Introduction: Presents the contextualization of the topic in a logical sequence. Explain the research problem's importance and justify the need to carry out the study. The contextualization of the problem can be improved using more current literature.

Done based in your valuable comment

2

Line 66-74

Line 83-88

Methodology: The psychometric characteristics of the scales used in this sample should be included. Clarify the scales used have been validated for the population of Saudi Arabia?

The scales used were not validated in Saudi Arabia; however, the scales were validated in other previous studies outside Saudi Arabia.

FOC questionnaire, The total scale revealed high reliability (r= 0.84), as reported by Sanjari et al. [24].

For Multidimensional Scale of Perceived Social Support (MSPSS).

" The Cronbach's α coefficient for family, friends, and significant other subscales is 0.87, 0.85, and 0.91, respectively. The total scale reliability is 0.88, as reported by Zimet et al. [25]. "

4

Materials and methods

Line 173

Line 179-181

Results: The results are consistent with the study objectives and the method. The data presented is sufficient and allows the authors to develop the proposed analysis.

Thank you very much for your comments and suggestions to improve our manuscript. We considered them in the revised version.

-

-

Discussion: The results were discussed with other authors, highlighting similarities and discrepancies regarding the knowledge already published in the international literature. However, the types of fears and their implications could be explored in greater detail, not just the differences between nulliparous and multiparous women.

Done based on your valuable comments

10

Discussion section

Line 307-337.

It would also be interesting if the authors could explain the implications of the results for clinical practice more clearly. In particular, it identifies strategies for adequately managing fear of childbirth and anticipating adverse outcomes.

Added based on your valuable comment.

12

Practical implications and further studies.

Line 408-417

Conclusion: They should also consider revising the findings to contribute more meaning to existing knowledge in the area.

Modified based on your valuable comment.

12

Conclusion

Line 4019-423.

References: Authors should update some of the references used in the manuscript to include more recent research. About half of the references are more than five years old.

New references were added based in your valuable comment.  

13

References

lines

458-462.

468-469

523-527.

Reviewer 2 Report

Comments and Suggestions for Authors

The paper titled “Determinant of Prenatal Childbirth Fear during the Third Trimester among Low-risk Expectant Mothers" is generally good, but some questions require clarification:

Title:

Please write the type of this study at the end of the title.

Abstract

-  Please rewrite the Objectives of the abstract, it is not clear enough.

Introduction:

-Introduction is good

Methods:

-Mention the way of participant allocation in this study? Randomly or not?

-Mention exclusion criteria

Results:

-write significance on Figure 2.

Discussion:

Line 312 ….Additionally, our findings indicated that primiparous and multiparous women with good families and spouse support were less likely to have FOC. In addition, Primiparous 313 women with low friends support had a higher probability of significant FOC…. Please explain.

-please rewrite Conclusion in obvious manner.

Reference:
-need to be checked

- Overall:

There are a few grammar errors that need to be checked.

Comments on the Quality of English Language

Moderate editing of the English language required

There are a few grammar errors that need to be checked.

Author Response

Reviewer  2 comments

We are very grateful to you for sending us your comments on our manuscript; according to these constructive comments provided by you, we have carefully revised the article as described below. They have significantly improved the quality and value of our manuscript. We hope you will find it to be a high-quality scientific work compatible with the standards of the healthcare journal. Please note all reviewer (2) comments are highlighted in green. The revisions are as follows: 

Comment

Authors response

Page

Paragraph number

The paper titled “Determinant of Prenatal Childbirth Fear during the Third Trimester among Low-risk Expectant Mothers" is generally good, but some questions require clarification:

Thank you very much for your comments and suggestions to improve our manuscript. We considered them in the revised version.

-

-

Title:

-         

Please write the type of this study at the end of the title.

Added based on your valuable comment.

1

Title

Line 3,4

Abstract

-  Please rewrite the Objectives of the abstract, it is not clear enough.

Added based on your valuable comment.

1

Abstract

Line 12,13

Introduction:

-Introduction is good

Thank you very much for your comments and suggestions to improve our manuscript. We considered them in the revised version.

Methods:

-Mention the way of participant allocation in this study? Randomly or not?

Clarified based on your valuable comment

3,4

Sampling technique

Line 143-154.

-Mention exclusion criteria

Clarified based on your valuable comment

3

Participants

Line 133-136.

Results:

-write significance on Figure 2.

Done based on your valuable comment

8

Results

255-257.

Discussion:

Line 312 ….Additionally, our findings indicated that primiparous and multiparous women with good families and spouse support were less likely to have FOC. In addition, Primiparous 313 women with low friends support had a higher probability of significant FOC…. Please explain.

The Multidimensional Scale of Perceived Social Support (MSPSS) is composed of three subscales to evaluate the social support provided by the family (4 items), friends (4 items), and significant others (4 items).

The results of the current study reported that the family social support subscale scale and significant others (spouse) subscale scale were significantly associated with FOC among both nulliparous and multiparous women.

Friends support subscale scale was significantly associated with FOC among nulliparous women only.

The description of the scale is provided on page 4

Data collection tools

Lines 174-181.

-please rewrite Conclusion in obvious manner.

Done based on your valuable comment

12

Conclusion

Lines 418-422.

Reference:

-need to be checked

Done

12-14

References

Lines 440-547.

- Overall:

There are a few grammar errors that need to be checked.

The manuscript was rechecked for grammar errors using Grammarly premium account and report is available on request.

-

-

Reviewer 3 Report

Comments and Suggestions for Authors

I have thoroughly reviewed your paper and appreciate your efforts. I've given constructive feedback to improve your article's quality. Please consider my suggestions to enhance clarity and impact. Once you make revisions, I will gladly review the updated version. Your commitment to improvement is commendable, and I look forward to seeing how your article evolves.

Best regards.

Abstract

  1. To increase readability, divide it into sections such as background, methods, results, and conclusions.
  2. In scientific writings it is important to avoid the use of abbreviations or acronyms without first defining them. For example, "FOC" (fear of childbirth) should be spelled the first time it appears in the summary.
  3. Clarify the purpose of the study by specifying the main research question or hypothesis to provide a clearer focus for readers..

 Introduction

       Consider the structure of the introduction with separate paragraphs for different aspects of the introduction, such as the importance of childbirth, psychological and physiological aspects, the concept of fear of childbirth (FOC) in lagaii.

       Avoid using acronyms or abbreviations without prior definition. For example, "FOC" (fear of childbirth)

       Some sentences are very long and complicated. Break them up into smaller, more manageable sentences..

Material and Methods

1.     For better organization, divide the Materials and Methods section into subsections such as study setting and participants, sample size and sampling technique, data collection tools, data collection methods, ethical approvals, and statistical analysis and analysis divided.

2.     Define abbreviations or acronyms before using them in the text.

3.     Include appropriate references from the scales mentioned above, such as the FOC and MSPSS questionnaires.

4.     Clarify the criteria for “significant FOC” and provide references if the term is derived from valid literature.

RESULTS

a.     Clarity and consistency: The results section provides valuable data, but there are some improvement suggestions to improve clarity and consistency:

b.     Ensure consistent formatting of numbers and percentages throughout the section. For example, use "50.1%" instead of 50.1% and "n=160" instead of "n=(160)".

c.     Ensuring consistency in measurement units. For example, when entering "age", specify whether it is in years.

d.     Check for typographical errors and inconsistencies in data presentation. For example, there is a missing value in the "Number of pars (times)" section.

e.     Consider using subheadings to divide the results section into subsections so that readers can more easily navigate and understand the information.

f.      Avoid overloading the text with statistical values and significance levels. Use tables and figures to present data more fully.

g.     Consider using visual aids such as charts or graphs to illustrate key findings, especially when comparing different groups..

Discussion:

       Discussion provides a comprehensive analysis of the study findings. However, to increase readability and clarity, consider dividing it into subsections with clear headings to address different aspects of the discussion. For example, one could have subsections for FOC members, FOC-related factors, and the role of social support in FOC.

       Providing an interpretation of the concepts of the results and their messages. Explain why primiparous women may be more concerned about FOC compared to multiparous women. Discuss the psychological and emotional factors that may contribute to this difference.

       While referencing previous studies, further present and discuss similarities and differences between your research and previous research. What insight or contribution does your study make to the existing literature on FOC?

       Verify that your study shows an association between some specific factor and significant FOC, but does not establish causation. Explain the relationships for inferring causality from cross-sectional data.

       Discuss the practical implications of your findings. How can health care providers, particularly nurses and midwives, use this information to improve the care and support provided to pregnant women? What strategies or interventions might be effective in reducing FOC, especially among high-risk groups?

       State clearly the limitations of your study, such as the possibility of bias, cross-sectional, and the fact that some relevant factors were not assessed. Acknowledging these correlations will help readers interpret the results correctly.

       Suggest directions for future research in this field. Are there specific areas or aspects related to FOC that require further investigation? Based on your findings, pose possible research questions or hypotheses..

Comments on the Quality of English Language

I have thoroughly reviewed your paper and appreciate your efforts. I've given constructive feedback to improve your article's quality. Please consider my suggestions to enhance clarity and impact. Once you make revisions, I will gladly review the updated version. Your commitment to improvement is commendable, and I look forward to seeing how your article evolves.

Best regards.

Abstract

  1. To increase readability, divide it into sections such as background, methods, results, and conclusions.
  2. In scientific writings it is important to avoid the use of abbreviations or acronyms without first defining them. For example, "FOC" (fear of childbirth) should be spelled the first time it appears in the summary.
  3. Clarify the purpose of the study by specifying the main research question or hypothesis to provide a clearer focus for readers..

 Introduction

       Consider the structure of the introduction with separate paragraphs for different aspects of the introduction, such as the importance of childbirth, psychological and physiological aspects, the concept of fear of childbirth (FOC) in lagaii.

       Avoid using acronyms or abbreviations without prior definition. For example, "FOC" (fear of childbirth)

       Some sentences are very long and complicated. Break them up into smaller, more manageable sentences..

Material and Methods

1.     For better organization, divide the Materials and Methods section into subsections such as study setting and participants, sample size and sampling technique, data collection tools, data collection methods, ethical approvals, and statistical analysis and analysis divided.

2.     Define abbreviations or acronyms before using them in the text.

3.     Include appropriate references from the scales mentioned above, such as the FOC and MSPSS questionnaires.

4.     Clarify the criteria for “significant FOC” and provide references if the term is derived from valid literature.

RESULTS

a.     Clarity and consistency: The results section provides valuable data, but there are some improvement suggestions to improve clarity and consistency:

b.     Ensure consistent formatting of numbers and percentages throughout the section. For example, use "50.1%" instead of 50.1% and "n=160" instead of "n=(160)".

c.     Ensuring consistency in measurement units. For example, when entering "age", specify whether it is in years.

d.     Check for typographical errors and inconsistencies in data presentation. For example, there is a missing value in the "Number of pars (times)" section.

e.     Consider using subheadings to divide the results section into subsections so that readers can more easily navigate and understand the information.

f.      Avoid overloading the text with statistical values and significance levels. Use tables and figures to present data more fully.

g.     Consider using visual aids such as charts or graphs to illustrate key findings, especially when comparing different groups..

Discussion:

       Discussion provides a comprehensive analysis of the study findings. However, to increase readability and clarity, consider dividing it into subsections with clear headings to address different aspects of the discussion. For example, one could have subsections for FOC members, FOC-related factors, and the role of social support in FOC.

       Providing an interpretation of the concepts of the results and their messages. Explain why primiparous women may be more concerned about FOC compared to multiparous women. Discuss the psychological and emotional factors that may contribute to this difference.

       While referencing previous studies, further present and discuss similarities and differences between your research and previous research. What insight or contribution does your study make to the existing literature on FOC?

       Verify that your study shows an association between some specific factor and significant FOC, but does not establish causation. Explain the relationships for inferring causality from cross-sectional data.

       Discuss the practical implications of your findings. How can health care providers, particularly nurses and midwives, use this information to improve the care and support provided to pregnant women? What strategies or interventions might be effective in reducing FOC, especially among high-risk groups?

       State clearly the limitations of your study, such as the possibility of bias, cross-sectional, and the fact that some relevant factors were not assessed. Acknowledging these correlations will help readers interpret the results correctly.

       Suggest directions for future research in this field. Are there specific areas or aspects related to FOC that require further investigation? Based on your findings, pose possible research questions or hypotheses..

Author Response

Reviewer 3 comments

We are very grateful to you for sending us your comments on our manuscript; according to these constructive comments provided by you, we have carefully revised the article as described below. It has significantly improved the quality and value of our manuscript. We hope you will find it to be a high-quality scientific work compatible with the standards of the healthcare journal. Please note all reviewer (3) comments are highlighted in yellow. The revisions are as follows: 

Comment

Authors response

Page

Paragraph number

Abstract

  1. To increase readability, divide it into sections such as background, methods, results, and conclusions.

Done 

We make it plodded

1

Abstract

Lines: 10-30.

  1. In scientific writings it is important to avoid the use of abbreviations or acronyms without first defining them. For example, "FOC" (fear of childbirth) should be spelled the first time it appears in the summary.

Done (The full name of the abbreviations was written when they first appeared)

1

All over the document

  1. Clarify the purpose of the study by specifying the main research question or hypothesis to provide a clearer focus for readers.

The aim of the study is presented at the end of the background section in the abstract. 

We highlighted it

1

Abstract

Lines: 12-13.

 Introduction

• Consider the structure of the introduction with separate paragraphs for different aspects of the introduction, such as the importance of childbirth, psychological and physiological aspects, the concept of fear of childbirth (FOC) in lagaii.

Done all over the introduction

1-3

Introduction

Lines: 34-116

•       Avoid using acronyms or abbreviations without prior definition. For example, "FOC" (fear of childbirth)

Checked all over the whole document

-

all over the document

•       Some sentences are very long and complicated. Break them up into smaller, more manageable sentences.

Done all over the introduction

1-2

all over the introduction

Material and Methods

1.     For better organization, divide the Materials and Methods section into subsections such as study setting and participants, sample size and sampling technique, data collection tools, data collection methods, ethical approvals, and statistical analysis and analysis divided.

Done

3-5

All over the Material and Methods

2.     Define abbreviations or acronyms before using them in the text.

Done

-

All over the document

3.     Include appropriate references from the scales mentioned above, such as the FOC and MSPSS questionnaires.

Done

4

Lines: 163,175.

4.     Clarify the criteria for “significant FOC” and provide references if the term is derived from valid literature.

The criteria were clarified in the operational definition.

Operational definition: significant FOC is a type of FOC that requires medical help to prevent potential drawbacks. In the current study, it was assessed using an FOC questionnaire, where the higher total scale score (31 to 60) indicates the presence of significant FOC .

3

Materials and methods

Lines: 118-121.

RESULTS

a.     Clarity and consistency: The results section provides valuable data, but there are some improvement suggestions to improve clarity and consistency:

Thank you very much for your comments and suggestions to improve our manuscript. We considered them in the revised version.

-

-

b.     Ensure consistent formatting of numbers and percentages throughout the section. For example, use "50.1%" instead of 50.1% and "n=160" instead of "n=(160)".

The result section was carefully revised based on your valuable comments; however, it is difficult to put or remove the brackets according to sentence structure in some sentences.

"n=160" instead of "n=(160)".    Done

6-9

Results section

c.     Ensuring consistency in measurement units. For example, when entering "age", specify whether it is in years.

Done all over the results

6-9

Results section

d.     Check for typographical errors and inconsistencies in data presentation. For example, there is a missing value in the "Number of pars (times)" section.

The results section was carefully revised based on your valuable comments. However, in Table 3, there are some variables that are not applicable for nulliparous women, such as previous labor-related complications.

6-9

Results

e.     Consider using subheadings to divide the results section into subsections so that readers can more easily navigate and understand the information.

Done all over the results

6-9

Results

f.      Avoid overloading the text with statistical values and significance levels. Use tables and figures to present data more fully.

Done all over the results

6-9

Results

g.     Consider using visual aids such as charts or graphs to illustrate key findings, especially when comparing different groups.

We agree with your valuable comment. We compared the total fear of childbirth among the two groups using chart.

The other tables comparing means and determinants of fear of childbirth are difficult to represent using charts.

8

Figure 2

Discussion:

•       Discussion provides a comprehensive analysis of the study findings. However, to increase readability and clarity, consider dividing it into subsections with clear headings to address different aspects of the discussion. For example, one could have subsections for FOC members, FOC-related factors, and the role of social support in FOC.

Done all over the discussion

9-11

Discussion section

•       Providing an interpretation of the concepts of the results and their messages. Explain why primiparous women may be more concerned about FOC compared to multiparous women. Discuss the psychological and emotional factors that may contribute to this difference.

Done all over discussion

9-11

Discussion section

•       While referencing previous studies, further present and discuss similarities and differences between your research and previous research. What insight or contribution does your study make to the existing literature on FOC?

Clarified at the end of the discussion section

11

Discussion

Line 390-396.

•       Verify that your study shows an association between some specific factor and significant FOC, but does not establish causation. Explain the relationships for inferring causality from cross-sectional data.

Added in the Strengths and limitations of the study

11,12

Strengths and limitations of the study

Lines: 397-407

•       Discuss the practical implications of your findings. How can healthcare providers, particularly nurses and midwives, use this information to improve the care and support provided to pregnant women? What strategies or interventions might be effective in reducing FOC, especially among high-risk groups?

Added based on your valuable comment.

12

Practical implications and further studies.

Lines: 408-4017

•       State clearly the limitations of your study, such as the possibility of bias, cross-sectional, and the fact that some relevant factors were not assessed. Acknowledging these correlations will help readers interpret the results correctly.

Added based on your valuable comment.

11,12

Strengths and limitations of the study

 Lines: 397-407

•       Suggest directions for future research in this field. Are there specific areas or aspects related to FOC that require further investigation? Based on your findings, pose possible research questions or hypotheses..

Added based on your valuable comment.

12

 further studies.

Lines: 415-417.

Round 2

Reviewer 1 Report

Comments and Suggestions for Authors

 Dear authors

I consider that they made the suggested improvements, which improved the clarity and scientific quality of the manuscript. We recommend accepting the article for publication.

We recommend accepting the article for publication.

Reviewer 2 Report

Comments and Suggestions for Authors

Thank you for response

Reviewer 3 Report

Comments and Suggestions for Authors

The revisions to your paper have been diligently implemented, resulting in a fully corrected and now acceptable manuscript. We extend our gratitude for your dedicated efforts and collaborative approach in elevating the overall quality of your article.

We wish you continued success in your forthcoming research and writing endeavors.